# A Facile Fabrication of CdSe/ZnS QDs—Block Copolymer Brushes-Modified Graphene Oxide Nanohybrid with Temperature-Responsive Behavior

**DOI:** 10.3390/ma15093356

**Published:** 2022-05-07

**Authors:** Yajiao Song, Hongcui Yu, Xiaohui Wang, Jinglin Liu, Jinghai Liu

**Affiliations:** 1College of Chemistry and Materials Science, Inner Mongolia Minzu University, Tongliao 028000, China; yajijaotzjz@163.com (Y.S.); yuhongcui526@163.com (H.Y.); wxh1009@126.com (X.W.); 2Key Laboratory of Natural Products Chemistry and Functional Molecular Synthesis, Inner Mongolia Minzu University, Tongliao 028000, China; 3Inner Mongolia Key Laboratory of Carbon Nanomaterials, Inner Mongolia Minzu University, Tongliao 028000, China

**Keywords:** graphene oxide, nanohybrid, thermo-responsive

## Abstract

In this paper, we described a straightforward one-step chemical method for the synthesis of semiconductor quantum dots(QDs)—block copolymer brushes functionalized graphene oxide(GO) fluorescence nanohybrids. The azobenzene-terminated block copolymer poly(N-isopropylacrylamid)-b-poly(styrene-co-5-(2-methacryoylethyloxymethyl)-8-quinolinol)(PNIPAM-*b*-P(St-*co*-MQ)) was modified on the surface of GO sheets via host–guest interactions between β-cyclodextrin-modified GO and azobenzene moieties, and simultaneously CdSe/ZnS QDs were integrated on the block copolymer brushes through the coordination between 8-hydroxyquinoline units in the polymer brushes and CdSe/ZnS QDs. The resulting fluorescence nanohybrid exhibited dual photoluminescence at 620 nm and 526 nm, respectively, upon excitation at 380 nm and LCST-type thermo-responsive behavior which originated from the change in the PNIPAM conformation in the block copolymer brushes of GO sheets.

## 1. Introduction

Graphene, as the youngest two-dimensional nanomaterial, sparked interest in the field of materials science following the successful fabrication of single-atom-thick graphene [1,2]. Graphene oxide (GO), as a derivative of graphene, which is easily obtained from graphite, has also gained considerable attention because of its abundant oxygen-containing functional groups on the basal plane and the edges of graphene [1,2,3,4,5,6,7,8,9]. These various oxygen-containing organic moieties such as carboxyl, hydroxyl, and epoxy groups provide GO with excellent water dispersibility [4]. Furthermore, the presence of these reactive groups makes GO highly attractive for easy tailoring and for the development of new advanced materials [4,6].

Recently, the concept of “organic/inorganic nano-composites” has been extended to the field of polymer–GO hybrids and has achieved success. Numerous effective methods for the fabrication of polymer–GO hybrids have been reported in the past few years [10,11,12,13,14,15,16,17,18,19,20]. In general, the fabrication techniques of polymer-functionalized GO include both covalent and non-covalent strategies. In the covalent routes, “grafting from” and “grafting to” are two common strategies with which to grow polymer brushes on the surface of GO [10]. The “grafting from” techniques which use initiator-functionalized GO as a precursor for polymerization include atom transfer radical polymerization (ATRP) and reversible addition-fragmentation chain transfer polymerization (RAFT) [11,12]. According to the “grafting to” methods, polymer brushes are directly covalent on the surface of a GO sheet. Among “grafting to” methods, esterification [13], click chemistry [14], nitrene chemistry [15,16], amidation [17] and radical addition [18] are very common. Besides, non-covalent modifications such as H-bonding [19] and the π-π stacking interaction on GO surfaces [20] have also been viewed as effective methods to prepare polymer–GO hybrids. In short, all the above methods have their own advantages and shortcoming. For instance, in “grafting to” methods, well-defined polymers are able to modify GO, but this is not the case with “grafting from” techniques because of the serious steric hindrance from the attached polymer brushes [21]. Conversely, “grafting from” methods provide GO hybrids with a high grafting density and rough control over composition, but multi-step syntheses are ordinarily necessary [21,22]. When a comparison is made between covalent and non-covalent routes, it is found that the covalent attachment has the potential of disrupting the conjugated structure of GO, leading to compromised physical properties of GO. However, the modification of GO using non-covalent techniques could retain the conjugated structure of GO after modification [12,23]. 

Herein, we designed and synthesized thermo-responsive block copolymer brushes and semiconductor quantum dots functionalized graphene oxide fluorescence nanohybrids using a straightforward one-step chemical method. Firstly, the azobenzene-terminated block-copolymer-brushed PNIPAM-*b*-P(St-*co*-MQ) was prepared via reversible addition-fragmentation chain transfer polymerization (RAFT) (see Figure 1). The block copolymer brush were designed to contain PNIPAM as a thermo-responsive segment and MQ units as a ligand which could coordinate with semiconductor quantum dots as a fluorescent component. The block copolymer brush PNIPAM-*b*-P(St-*co*-MQ) was functionalized on the surface of GO sheets via host–guest interactions between β-cyclodextrin-modified GO and azobenzene moieties. At the same time, CdSe/ZnS QDs were integrated on the block copolymer brushes through the coordination between 8-hydroxyquinoline units and CdSe/ZnS QDs. Furthermore, the resulting fluorescence nanohybrid possessed an LCST-type thermo-responsive behavior originating from the change in the PNIPAM conformation in the block copolymer of a single-atom thickness GO (see Figure 2).

## 2. Experiment Section

### 2.1. Chemicals

N-isopropylacrylamide (NIPAM, 99%, Aldrich, St. Louis, MO, USA) was recrystallized in hexane before use. Then, 2,2-Azoisobutyronitrile (AIBN, Macklin, Shanghai, China) was recrystallized from ethanol (95%). CdSe/ZnS QDs (4.17 mg·mL^−1^) were purchased from Xingshuo Nanotechnology Co., Ltd. (Suzhou, China) Graphene oxide [24], 2-[(dodecylsulfanyl) carbonothioylsulfanyl] propanoic acid (RAFT agent) [25], 5-(2-methacryloylethyloxymethyl)-8-quinolinol (MQ) [26] and mono-[6-(2-aminoethy-lamino)-6-deoxy]-cyclodextrin(6-NH_2_-β-CD) [27] were synthesized according to procedures detailed in the literature. All other reagents were purchased from Xilong Chemical Reagent Co. (Shantou, China) and used as received. 

### 2.2. Synthesis of the Azobenzene-Terminated Block Copolymer PNIPAM-b-P(St-co-MQ) 

In order to synthesize azobenzene-terminated block copolymer PNIPAM-*b*-P(St-*co*-MQ), we prepared a novel azobenzene functionalized chain transfer agent (azobenzene–RAFT agent) as follows [27]: Briefly, 80 mL of dry dichloromethane was loaded into a 100 mL round-bottom flask, followed by the addition of 0.36 g of the RAFT agent and 0.2 g 4-phenylazophenol. After stirring for 1 h in an ice bath with nitrogen, the dichloromethane solution of DCC (0.42 g) and 4-(N,N-dimethylamino) pyridine (DMAP) (0.24 g) was added into the mixture and the reaction was carried out at room temperature for 24 h. After that, the filtrate was collected and concentrated, dissolved in chloroform and washed with water several times. Then, the organic layer was filtered and concentrated again after drying with MgSO_4_ overnight.

A typical synthesis of the azobenzene-terminated block copolymer was carried out as follows: NIPAM (2 g), AIBN (4 mg) and azobenzene–RAFT agent (4 mg) in a freshly distilled tetrahydrofuran solution (THF, 6 mL) were placed in a three-necked round-bottom flask. After three freeze–pump–thaw cycles, the above solution was heated gradually and was kept at 75 °C for 12 h under vacuum. The resulting product was centrifuged and washed with diethyl ether repeatedly. After drying in a vacuum oven overnight at room temperature, the copolymer PNIPAM was obtained. 

MQ (140 mg), AIBN (8 mg) and PNIPAM (600 mg) were added into the freshly distilled THF (6 mL) and stirred until the above solids were completely dissolved. After degassing with three freeze–pump–thaw cycles, the above solution was subsequently immersed into an oil bath and heated gradually to 75 °C for 12 h. Then, the resulting product was centrifuged and washed with diethyl ether repeatedly. After drying in a vacuum oven overnight at room temperature, the azobenzene-terminated block copolymer PNIPAM-*b*-P(St-*co*-MQ) was obtained. 

### 2.3. Synthesis of β-Cyclodextrin Modified GO(CD-GO)

The β-cyclodextrin-modified GO(CD-GO) hybrid was synthesized as follows [26]: NaOH (5.0 g), chloroacetic acid (5.0 g), GO (0.1 g) and deionized water (100 mL) were added to a 250 mL round-bottom flask and subjected to an ultrasonic bath for 2 h. The above products were centrifuged and washed with deionized water and then dried in vacuo for 24 h. After that, the above products (0.1 g) were dispersed in 100 mL deionized water followed the addition of 1-ethyl-3-(3-dimethylaminopropyl) carbodiimide hydrochloride (EDC, 115 mg) and N-Hydroxysulfosuccinimide (NHS, 60 mg). After ultrasonication for 30 min at room temperature, 0.2 g 6-NH_2_-β-CD was added, followed by an additional reaction for 24 h. The resulting product was centrifuged and washed with deionized water, respectively. After drying in a vacuum oven overnight at room temperature, the CD-GO hybrid was obtained.

### 2.4. Synthesis of QDs/Polymer/GO Fluorescence Nanohybrid

A round-bottom flask containing 5 mL deionized water was charged with PNIPAM-*b*-P(St-*co*-MQ) (30 mg), CD-GO (10 mg) and CdSe/ZnS QDs (20 μL). After ultrasonication for 30 min, the reaction was performed at room temperature for 36 h, then the above mixture was centrifuged and washed with deionized water several times. The resulting QDs/polymer/GO fluorescence nanohybrid was obtained after drying in vacuum for 12 h.

### 2.5. Monitoring Thermo-Response Behavior 

The resulting QDs/polymer/GO fluorescence nanohybrid (10 mg) and deionized water (20 mL) were added into a 50 mL beaker and subjected to an ultrasonic bath for 0.5 h. After that, 5 mL of the above aqueous solution (0.5 mg·mL^−1^) was poured into the quartz cell (1 cm path length) and placed in the cell holder of the spectrophotometer. The fluorescent signals were collected after having stabilized for 5 min at different temperatures (27~51 °C).

### 2.6. Measurements

The ^1^H NMR spectra were recorded on a Bruker Avance III 600 spectrometer (resonance frequency: 600 MHz, Brooke Technology Co., Ltd., Germany) using CDCl_3_ as a solvent. Gel permeation chromatography (GPC, Waters Corporation, Milford, MA, USA) measurements were performed at a flow rate of 1.0 mL min^−1^ at 25 °C with DMF as the eluent. Fourier transform infrared (FT-IR, Nicolet, USA) spectra were obtained using a Perkin-Elmer 2000 spectrometer (KBr disks). A JEOL 2000FX electron microscope was used to obtain the transmission electron microscopy (TEM, JEOL, Akishima, Japan) images. A TGA 2050 thermogravimetric (Polymer Laboratories, TGA1000, Church Stretton, UK) analyzer was used for the thermogravimetric analysis and the heating rate was 10 °C min^−^^1^ from 20 °C to 800 °C under a nitrogen atmosphere. A PerkinElmer LAMBDA 750 UV-vis (Shimadzu, Tokyo, Japan) spectrophotometer was used for the UV−vis absorption spectra studies (the range from 200 to 700 nm). The steady state photoluminescence spectroscopy was measured on a fluorescence spectrofluorometer (Edinburgh FLS980, UK) with an excitation at 380 nm (in a range from 400 to 700 nm).

## 3. Results and Discussion

### 3.1. Characterization of QDs/Polymer/GO Fluorescence Nanohybrid

In this work, we developed a semiconductor quantum dots—block copolymer brushes—functionalized GO fluorescence nanohybrid with a straightforward one-step chemical method. Figure 1 shows our key strategy for the fabrication of azobenzene-terminated block copolymers. In the first place, the azobenzene modified 2-[(dodecylsulfanyl) carbonothioylsulfanyl] propanoic acid (azobenzene–RAFT agent) was synthesized by esterification. The azobenzene-terminated block copolymer PNIPAM-*b*-P(St-*co*-MQ) was prepared using RAFT polymerization based on the monomers of N-isopropylacrylamide (NIPAM), styrene(St), 5-(2-methacryloylethyloxymethyl)-8-quinolinol(MQ) and the azobenzene–RAFT agent. The block copolymer brushes were designed to contain PNIPAM as a thermo-responsive segment and MQ units as a ligand which could coordinate with semiconductor quantum dots as a fluorescent component. As shown in Figure 2, the azobenzene-terminated block copolymer brushes PNIPAM-*b*-P(St-*co*-MQ) were functionalized on the surface of GO sheets via host–guest interactions between β-cyclodextrin-modified GO and azobenzene moieties. Additionally, CdSe/ZnS QDs were simultaneously integrated on the block copolymer brushes through the coordination between 8-hydroxyquinoline units and CdSe/ZnS QDs. The resulting fluorescence nanohybrid exhibited dual photoluminescence at 620 nm and 526 nm, respectively, upon excitation at 380 nm and an LCST-type thermo-responsive behavior which originated from the change in the PNIPAM conformation in the block copolymer brushes of GO sheets. CdSe/ZnS QDs and the metaloquinolate complexes were highly favorable fluorescent donors and GO material can be used as an efficient Förster resonance energy transfer (FRET) acceptor in an optical sensor due to its high long-distance fluorescent quenching efficiency with a high signal-to-background ratio. In this article, the FRET between GO and CdSe/ZnS QDs moieties in block polymer brushes was successfully governed by the manipulation of conformational features of the PNIPAM chains that respond to temperature changes.

The ^1^H NMR spectra of the RAFT agent and azobenzene–RAFT agent in CDCl_3_ are shown in Figure 3. As seen in Figure 3a, the chemical shift located at 3.20 ppm for the RAFT agent is attributed to the proton on the methylene group near the trithiocarbonates. The signal at 1.66 ppm is assigned to the proton on two methyl groups near the carboxyl group. The signal at 1.60–1.11 ppm is assigned to the proton on the methyl group at the end of the RAFT agent [24]. When the RAFT agent was modified with azobenzene, some new signals appeared in the spectra of the azobenzene–RAFT agent. As seen Figure 3b, the signals at 7.22–7.27(d, 2H, azobenzene-Hα), 7.46–7.54(m, 3H, azobenzene-Hγ) and 7.87–7.96 ppm (m, 4H, azobenzene-H_β_) are attributable to the protons of azobenzene of the azobenzene–RAFT agent [27]. As expected, the ^1^H NMR spectra of the above chain transfer agents testify to the successful synthesis of the RAFT agent and Azo–RAFT agent. 

Figure 4 presents the ^1^H NMR spectra of PNIPAM and block copolymer PNIPAM-*b*-P(St-*co*-MQ). As shown in Figure 4a, the signal at 4.0 ppm for the PNIPAM is assigned to the O=CNH proton, which illustrates that the copolymer PNIPAM was prepared successfully [28], while some new signals appear in the spectra of PNIPAM-b-P(OEGMA-co-MQ) (see Figure 4b). The signals at 6.40–7.10 ppm are protons of the phenyl ring and the signals at 8.5 and 8.7 ppm are the protons of MQ [26]. In addition, it can be inferred by integrating the area calculation that the molar ratio of MQ and St in the polymer was about 1:10. According to Figure 4, the average molecular weights of PNIPAM and block polymer PNIPAM-*b*-P(St-*co*-MQ) were calculated to be about 20.1 and 29 kDa. As compared with the GPC results (see Table 1: M_n,GPC_ = 21.0 kDa and 30.1 kDa), the calculated result from the ^1^H NMR spectra was similar with the GPC results. So, the structure of the above copolymers that we synthesized in the study can be assigned to PNIPAM_177_ and PNIPAM_177_-b -P(St_0.91_ -co-MQ_0.09_)_75_.

FTIR can reveal the characteristic vibrations of PNIPAM-*b*-P(St-*co*-MQ) after polymerization. As shown in Figure 5 line a, the prominent peaks at 1646 and 1538 cm^−1^ were characteristic of the absorption vibration of the C=O bands (amide I and II band) and the others (2864, 2922 and 2968 cm^−1^) are related to the fundamental stretching vibration of -CH(CH_3_)_2_ [28]. After REFT polymerization (see Figure 3b), some new bands appeared at 1781, 2939, 2985, 3043 and 710 cm^−1^ in the spectrum of block copolymer PNIPAM-*b*-P(St-*co*-MQ). The prominent peak at 1781 cm^−1^ corresponds to the C=O vibration of MQ units [26] and the bands at 3043, 2985, 2939 and 710 cm^−1^ originate from the characteristic peaks of benzene in polystyrene units [29]. FTIR spectra analyses confirm that the copolymerization was successfully completed.

Raman spectra can provide evidence for the conjugation of PNIPAM-*b*-P(St-*co*-MQ) block copolymer brushes onto the GO surface. As presented in Figure 6, two prominent bands can be observed at 1337 and 1609 cm^−1^. The band at 1337 cm^−1^, which we named as the D band, is link to the vibration of carbon atoms with dangling bonds in the mussy GO sheets, which verified the existence of sp^3^ carbon in the GO sheets, while the prominent peak at 1609 cm^−1^, named the G band and relating to the E_2g_ mode of GO sheets, was assigned to the number of sp^2^ carbon atoms [28,29,30]. The G bands in the QDs/polymer/GO fluorescence nanohybrid exhibited a gradual blue shift from 1609 to 1604 cm^−1^. The blue shift may be attributed to the tethering of β-cyclodextrin and block copolymer brushes on GO sheets as they deteriorate the alternate pattern of single-double bonds within the sp^2^ carbon sheets. Generally, the intensity ratio of D and G bands (I_D_/I_G_) is the most common metric used to characterize the defect density of graphene. For our samples, the intensity ratio of D and G bands (I_D_/I_G_) of the GO was about 1.13, while that of the QDs/polymer/GO fluorescence nanohybrids enhanced to 1.16. This interesting phenomenon should be attributed to the gradual increase in the sp^3^ carbon structure after polymerization [28,29]. In addition, the size of sp^2^ carbon clusters of graphene oxide and the QDs/polymer/GO fluorescence nanohybrids could be calculated using Knights empirical formula, which is as follows [31]:L_a_ = 4.35/(I_D_/I_G_)(1)
where L_a_ is designated as the size of sp^2^ carbon clusters and I_D_/I_G_ is the intensity ratio between the D and G bands. According to the above formula, the L_a_ data of GO are 3.85 nm and for the fluorescence nanohybrid are 3.75 nm. The decreasing L_a_ data prove that the block polymer and QDs modified on the surface of GO result in a gradual reduction in the graphitic structure, which results in an increase in the I_D_/I_G_ ratio and a decrease in the size of the graphitic domains [28,29,30]. Therefore, the above Raman spectral analyses provide further support for our conclusion that the block copolymer was successfully grafted onto GO sheets.

The morphologies of GO, CdSe/ZnS QDs, CD-GO and QDs/polymer/GO fluorescence nanohybrids were characterized with the TEM technique. As depicted in Figure 7a, the pristine GO sheets wrinkled due to the abundant polar oxygen-containing functional groups on the basal plane and the edges of graphene [28]. According to the TEM images collected from the purchased CdSe/ZnS QDs (seen Figure 7b), the CdSe/ZnS QDs with a diameter of about 8–10 nm and its internal lattice fringes can be clearly observed, indicating that the quantum dot has a good lattice structure. After modification with β-cyclodextrin though amidation (see Figure 7c), the monolayer GO sheet evidently thickened and the wrinkles and folding disappeared, while the appearance of dark areas indicate that β-cyclodextrin was modified on the surface of GO [26]. After semiconductor quantum dots(QDs)—block copolymer brushes were functionalized on GO sheets (see Figure 7d), a thin homogeneous polymer layer on the surface of GO with black patches over the GO sheets was observed. Meanwhile, CdSe/ZnS QDs were wrapped up via the coordination interaction between the N-containing functional heterocyclic ligand in the block polymer brushes and quantum dots. Therefore, the TEM images show that the azobenzene-terminated block copolymer brushes were anchored onto the GO sheets and the introduction of CdSe/ZnS QDs in the block polymer was carried out successfully.

A TGA analysis was carried out to study the thermal stability of GO and QDs/polymer/GO nanohybrid. As presented in Figure 8 line a, the GO was found to have approximately 15 wt% weight loss below 150 °C which resulted from the volatilization of physically absorbed water [32]. As a result of the thermal decomposition of liable oxygen-containing groups on its π-stacked structure, GO experienced considerable mass loss (approximately 50 wt%, except water) in the range of 200–600 °C [28,29,30]. As seen in Figure 8 line b, the QDs/polymer/GO nanohybrid was found to experience approximately 85 wt% weight loss when the temperature was elevated from 200 °C to 490 °C. As shown in Figure 8 line c, the thermal stability of GO is evidently lower than that of block copolymer, which illustrates that the thermal stability of GO can be enhanced with block copolymer. Moreover, the host–guest interaction to some extent and the TGA curve of QDs/polymer/GO nanohybrid prove this conclusion. On the other hand, according to the TGA results and the following formula, the grafting density of block copolymer brushes could be calculated [33]:(2)$${\tilde{A}}_{mg}=\frac{M_{C}W_{F}}{M_{F}W_{C}}$$
where *M_F_* is the average molecular weight (*M_n_*) of modified copolymer (come from GPC); *M_C_* is the relative molar mass of the carbon atom; *W_C_* is the weight fraction of the block copolymer modified graphene backbone (not including grafted polymer); and *W_F_* is the weight fractions of the grafted polymer. Therefore, based on the TGA curves of GO and the QDs/polymer/GO nanohybrid as well as the GPC measurement, the grafting density of the block copolymer PNIPAM-*b*-P(St-*co*-MQ) on the surface of GO could be calculated to be 1.1 chains per 100 carbons in this work.

### 3.2. Optical Properties of QDs/Polymer/GO Fluorescence Nanohybrid

Figure 9 displays the UV-vis absorption spectra of pristine GO sheets and the QDs/polymer/GO fluorescence nanohybrid, in which the inset is the UV-vis absorption spectrum of CdSe/ZnS QDs. As shown in Figure 9 line a, GO experienced an absorption peak at 235 nm, which resulted from the π-π* transition of the aromatic C=C bond, and the absorption at 300 nm could be assigned to the n-π* transition of C=O bonds [34]. The GO sheet functionalized with block copolymer PNIPAM-*b*-P(St-*co*-MQ) and CdSe/ZnS QDs (see Figure 9 line b). As a result of the π-π* electron transition from the quinoline ring [35], a new absorption peak is observable at 249 nm, while another absorption peak at 352 nm corresponds to the metal–quinolate transition in the polymer chains [36]. Besides, in the spectra of the QDs/polymer/GO fluorescence nanohybrid, the typical characteristics of CdSe/ZnS QDs at 620 nm cannot be clearly observed because the characteristic absorption of the CdSe/ZnS QDs is too small compared with the absorption from the QDs–quinolate transition.

Figure 10 presents the fluorescence spectrum of CdSe/ZnS QDs and the QDs/polymer/GO fluorescence nanohybrid. As seen in Figure 10a, the CdSe/ZnS QDs have a distinct red-light emission at 620 nm as excited by 380 nm at room temperature. While for the CdSe/ZnS QDs coordinated with 8-hydroxyquinoline units contained in the Azo-terminated block polymer of the GO sheet, there are two obvious emission centers. The emission at 620 nm originates from the inherent luminescence of semiconductor quantum dots and another strong emission at 526 nm results from the surface-coordination emission between Zn^2+^ contained in CdSe/ZnS QDs and 8-hydroxyquinoline contained in the Azo-terminated block polymer of the GO sheet. As a result, the QDs/polymer/GO fluorescence nanohybrid we prepared in this work shows two-channel fluorescence emission.

When the QDs/polymer/GO fluorescence nanohybrids were dispersed in aqueous solution, a homogeneous stable dark solution was observed at ambient temperature in the absence of any specific dispersant. More remarkably, the resulting dispersion could remain stable, and no obvious precipitation occurred even after storage under ambient conditions for more than seven days. In this work, the thermo-response behavior of the QDs/polymer/GO fluorescence nanohybrid in water was investigated by monitoring the change in luminescence intensity as a function of temperature. As depicted in Figure 11a, the fluorescence emission of the QDs/polymer/GO fluorescence nanohybrid did not change when the temperature increased from 27 °C to 38 °C. However, when the fluorescence nanohybrid was slowly heated to 51 °C, a sharp decrease in the relative emission intensity could be observed in the spectra. Interestingly, this phenomenon was induced by the change in Förster resonance energy transfer (FRET) efficiency from the QDs to the GO sheet, which was caused by the conformation change of PNIPAM in response to different temperatures [16,37,38]. At temperatures below 38 °C (LCST), the polymer unit PNIPAM becomes hydrophilic and the polymer chains display expanded conformation in water, leading to an increase in the distance between the QDs and the GO sheet. Therefore, the FRET process between the QDs donor and GO acceptor was suppressed. In contrast, when the temperature is above LCST, the PNIPAM becomes hydrophobic and the distance between QDs and GO sheet decreases. It is worth noting that the emission was not completely quenched in this aqueous solution; the reason for this is that a sufficient distance between GO and QDs which were modified on the block polymer was still provided by the globule state of the PNIPAM chains. The reversibility of the thermo-response of the QDs/polymer/GO fluorescence nanohybrid was also discussed. As shown in Figure 11b, 10 continuous cycles of the efficient on–off switching behaviour occurred in the aqueous solution of the QDs/polymer/GO fluorescence hybrid. By cooling the sample solution, the quenched fluorescence could be recovered completely. When re-heated above LCST, the GO-based fluorescence nanohybrid displays a similar PL quenching efficiency. The above study indicates that there is a strong interaction between the thermo-responsive block copolymer and GO sheets, which induced the high stability observed in this experiment.

## 4. Conclusions

In this work, the Azo-terminated block polymer PNIPAM-*b*-P(St-*co*-MQ)—functionalized GO organic–inorganic nanohybrid was fabricated via the host–guest interaction between β-cyclodextrin-modified GO and the azobenzene group in the block copolymer PNIPAM-*b*-P(St-*co*-MQ). Additionally, the QDs/polymer/GO fluorescence nanohybrid was formed by the coordinated interaction between N-containing functional heterocyclic ligand and semiconductor quantum dots CdSe/ZnS QDs. Furthermore, the resulting fluorescence nanohybrid was an LCST-type thermo-responsive nano-composite which originated from the change in the PNIPAM conformation in the block copolymer of the GO sheet. Thus, because of the extensive applications of the single-atom-thick graphene oxide as a main material for a variety of copolymer and semiconductor quantum dots, the design developed here could be an effective strategy for producing designed semiconductor quantum dots—polymer brushes functionalized graphene oxide(GO) fluorescence nanohybrids for electronic, optical, catalysis, environmental and new energy fields.

## Figures and Tables

**Figure 1 materials-15-03356-f001:**
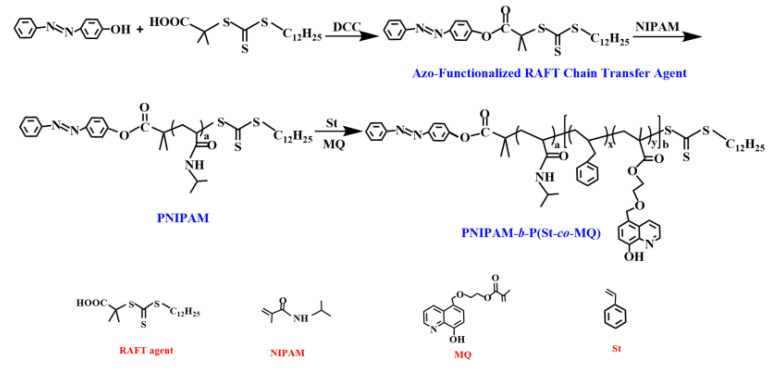
Schematic illustration for preparation of PNIPAM-*b*-(St-*co*-MQ).

**Figure 2 materials-15-03356-f002:**
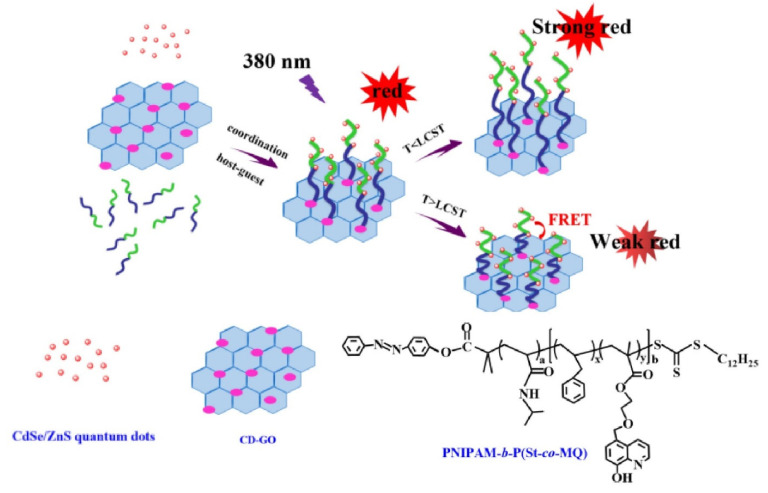
Schematic illustration of the preparation of QDs/polymer/GO and its thermo-response behavior.

**Figure 3 materials-15-03356-f003:**
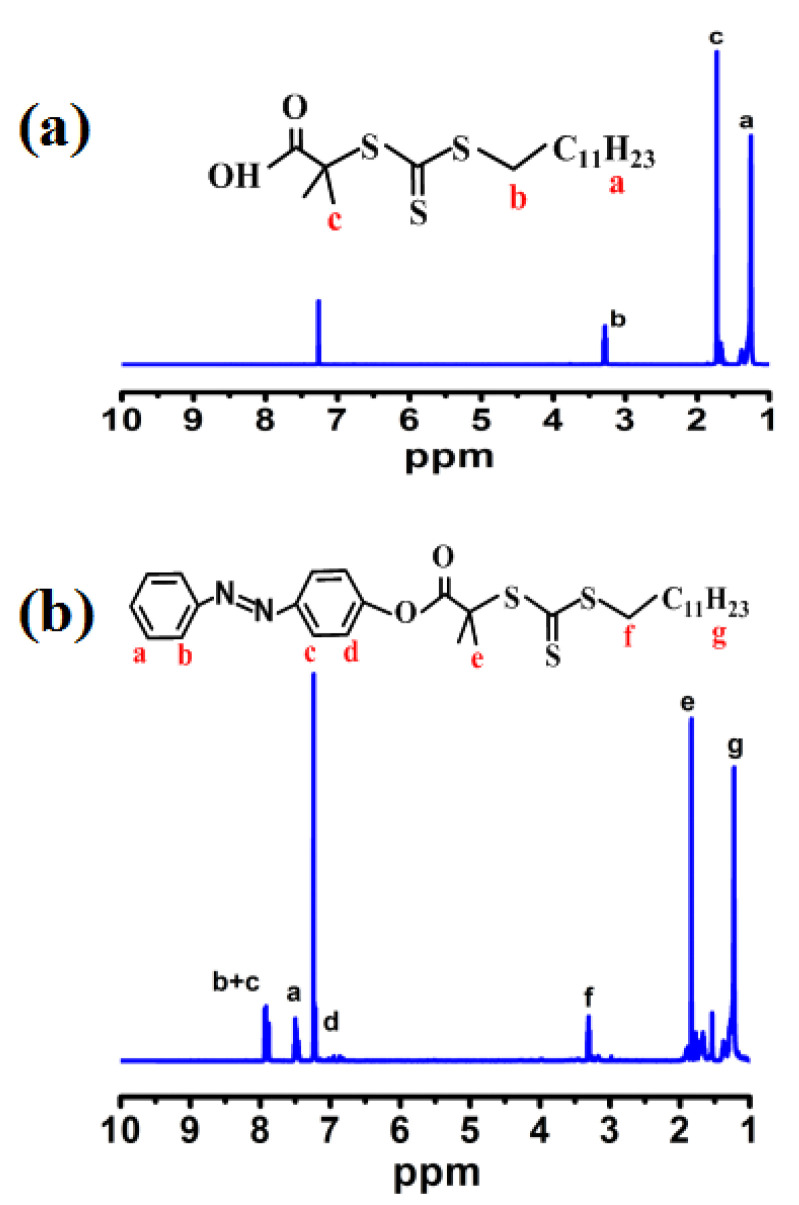
^1^H NMR spectra of RAFT (**a**) and azobenzene-RAFT (**b**).

**Figure 4 materials-15-03356-f004:**
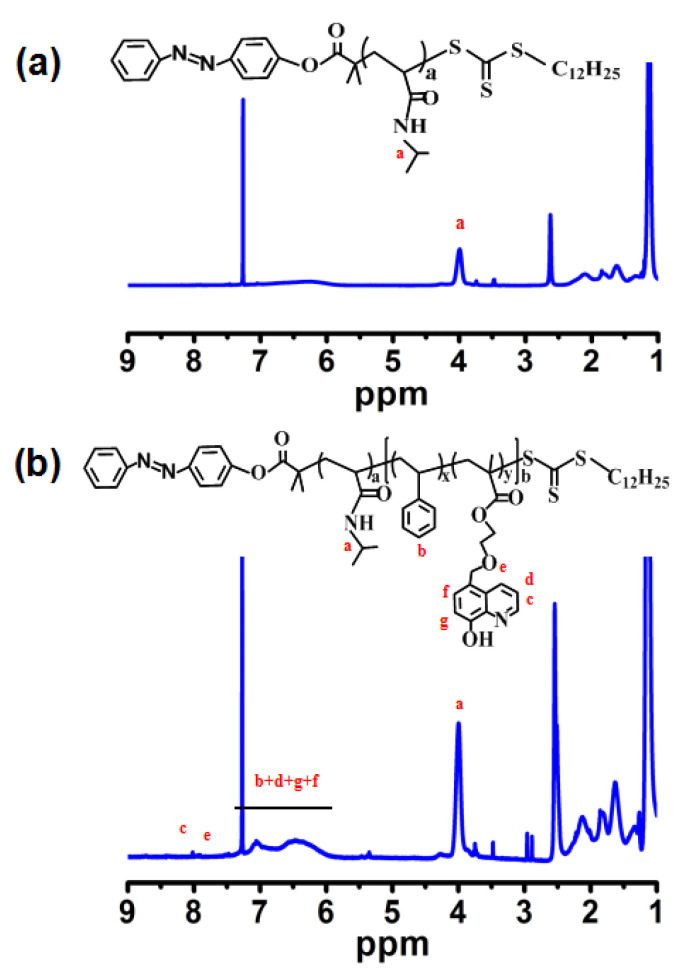
^1^H NMR spectra of PNIPAM (**a**) and PNIPAM-*b*-P(St-*co*-MQ) (**b**).

**Figure 5 materials-15-03356-f005:**
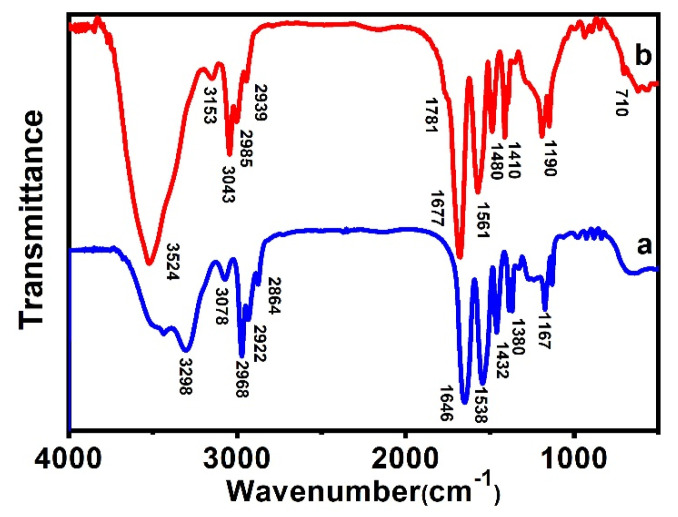
FT-IR spectra of PNIPAM (**a**) and block polymer PNIPAM-*b*-P(St-*co*-MQ) (**b**).

**Figure 6 materials-15-03356-f006:**
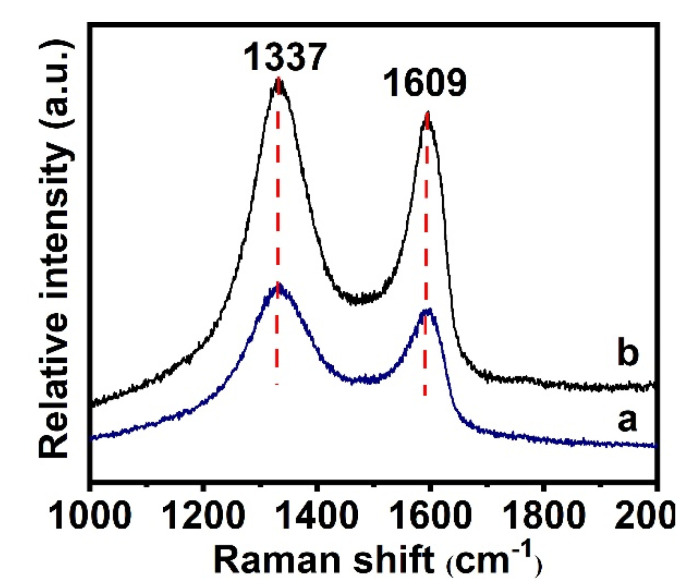
Raman spectra of GO (**a**) and QDs/polymer/GO fluorescence nanohybrid (**b**).

**Figure 7 materials-15-03356-f007:**
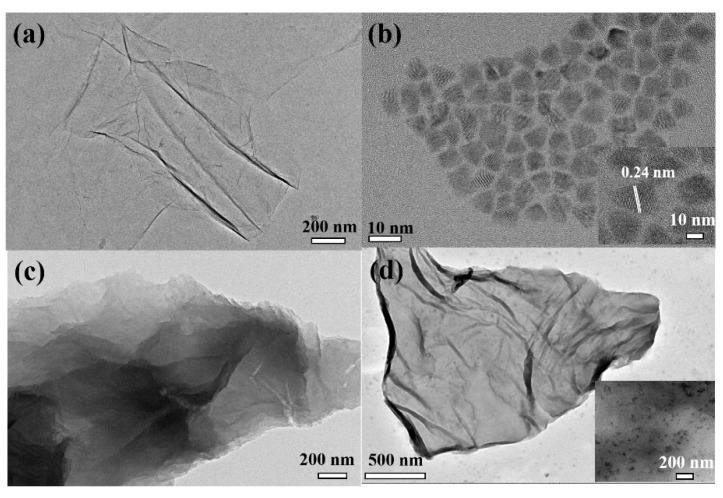
TEM micrograph of GO (**a**), CdSe/ZnS QDs (**b**), GO-CD (**c**) and fluorescence hybrid (**d**).

**Figure 8 materials-15-03356-f008:**
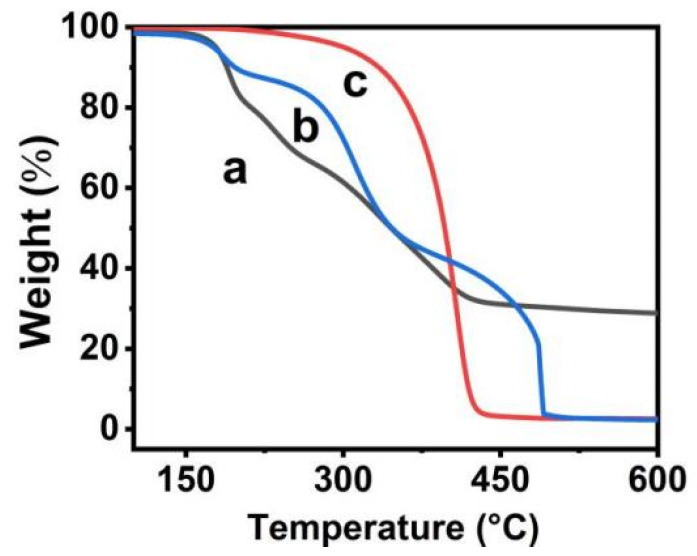
TGA spectra of GO (**a**), QDs/polymer/GO-fluorescence nanohybrid (**b**) and free block copolymer brushes (**c**) PNIPAM-*b*-P(St-*co*-MQ).

**Figure 9 materials-15-03356-f009:**
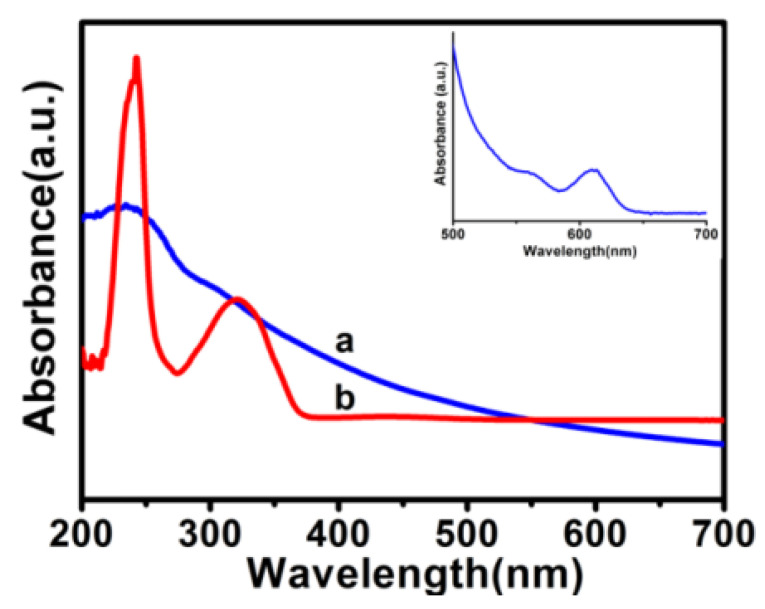
UV-vis absorption spectra of pristine GO (**a**), QDs/polymer/GO fluorescence nanohybrid (**b**) and CdSe/ZnS QDs (inset).

**Figure 10 materials-15-03356-f010:**
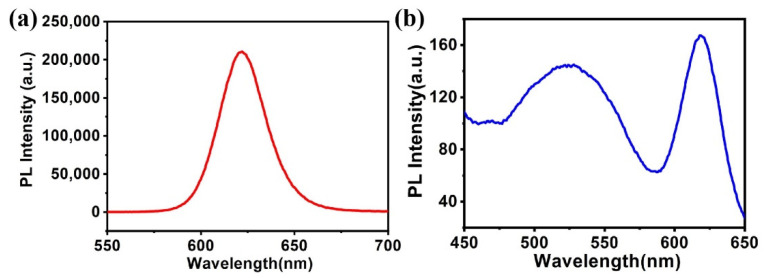
PL of CdSe/ZnS QDs (**a**) and QDs/polymer/GO fluorescence nanohybrid (**b**).

**Figure 11 materials-15-03356-f011:**
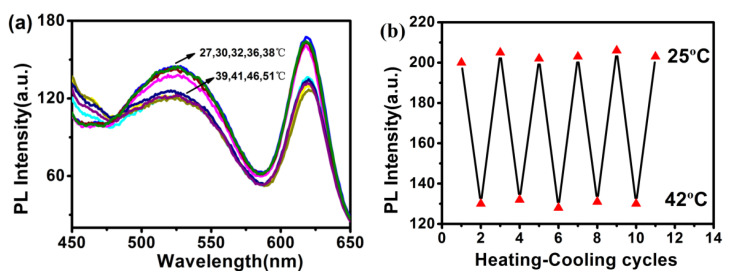
Evolution of fluorescence spectra as a function of temperature for QDs/polymer/GO fluorescence nanohybrid (**a**) and corresponding cycles of heating–cooling at above and below LCST of QDs/polymer/GO (**b**).

**Table 1 materials-15-03356-t001:** Characterization data of polymer synthesized by RAFT polymerization.

Polymer	M_NMR_ ^※^	M_n_	M_w_	PDI
PNIPAM_177_	20.1	21.0	32.0	1.37
PNIPAM_177_-*b*-P(St_0.91_-*co*-MQ_0.09_)_75_	29.0	30.1	45.5	1.37

^※^ M_NMR_ were calculated on the basis of ^1^H NMR results.

## Data Availability

The data presented in this study are available upon request from the corresponding author.

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
