# Peer review of "A Facile Fabrication of CdSe/ZnS QDs—Block Copolymer Brushes-Modified Graphene Oxide Nanohybrid with Temperature-Responsive Behavior"

_materials, 2022, doi:10.3390/ma15093356_

Round 1
Reviewer 1 Report
In this paper, the work done to prepare block copolymer PNIPAM-b-P(St-co-MQ) functionalized graphene oxide (GO) through coordinated interaction between heterocyclic ligand and semiconductor quantum dots. The steps taken for the preparation method were well defined. The results, analysis, and discussion were explained well, proving the fluorescence of the nanohybrid. The cited references were mixed from old and recent works.
English need a major revision, very seriously. Too many glaring grammatical, and structural errors, some space between words missing, repetitive word, etc… These just made the manuscript much challenging to read.
Line 10- Abstrsct to abstract
Abstract section needs to revise as it’s quite general and not summarize well the details content of the manuscript
MQ term is not well defined, especially in the first part of the draft
Last part of Introduction contains many details regarding experimental part and analysis, which not appropriate in this section.
Need to use term “Figure” instead of just mentioning “Scheme 1 and 2”
Some figure not properly aligned and mixed with the text.
In most part, the presentation was more on the acquired data. Discussion, especially in the mechanism of the formation in Scheme 1 and 2 was not properly addressed.
Reviewer 2 Report
- The manuscript is well written. However, the methodologies should be elaborated. Authors mentioned that the fluorescence intensity of the QDs/polymer/GO fluorescence nanohybrid with different temperatures in water was investigated. Therefore, the temperature and time (how long) should be clarified in methods.
- In “Results and discussion” section mostly, results are written without discussion.
- Application of this complex nanohybrid must be included (at least one).
Author Response
Point 1: The manuscript is well written. However, the methodologies should be elaborated. Authors mentioned that the fluorescence intensity of the QDs/polymer/GO fluorescence nanohybrid with different temperatures in water was investigated. Therefore, the temperature and time (how long) should be clarified in methods.
Response 1: The resulting QDs/polymer/GO fluorescence nanohybrid(10 mg) and deionized water(20 mL) were added into a 50 mL beaker and subjected to an ultrasonic bath for 0.5 h. After that 5 mL of the above aqueous solution (0.5 mg mL-1) was poured into the quartz cell (1 cm path length) followed placed in the cell holder of the spectrophotometer. The fluorescent signals were collected after having stabilized for 5 min at different temperature(27 ~51oC).
Point 2: 2.In “Results and discussion” section mostly, results are written without discussion.
Response 2: We thank the reviewer for the suggestion and improving the manuscripts. The errors have been corrected in the revised manuscript.
Point 3: Application of this complex nanohybrid must be included (at least one).
Response 3: The QDs/polymer/GO fluorescence nanohybrid fluorescence nanohybrids could be a robust temperature sensing platform because of its LCST-type thermo-responsive behavior which originated from the change in the PNIPAM conformation in the block copolymer brushes of GO sheets.
When the QDs/polymer/GO fluorescence nanohybrid were dispersed in aqueous solution, a homogeneous stable dark solution was observed at ambient temperature in the absence of any specific dispersant. More remarkably, the resulting dispersion could remain stable, and no obvious precipitation occurred even after storage under ambient conditions for more than seven days. In this work, the thermo-response behavior of the QDs/polymer/GO fluorescence nanohybrid in water was investigated by monitoring the change in luminescence intensity as a function of temperature. As depicted in Figure 11a, the fluorescence emission of QDs/polymer/GO fluorescence nanohybrid was not change with the temperature increased from 27 oC to 38 oC. However, when the fluorescence nanohybrid was heated to 51 oC slowly, a sharp decrease of the relative emission intensity could be observed in the spectra. Interestingly, this phenomenon was induced by the change in Förster resonance energy transfer (FRET) efficiency from the QDs to the GO sheet, which caused by the conformation change of PNIPAM in response to different temperature[16,37,38]. When temperature below 38 oC (LCST), the polymer units PNIPAM becomes hydrophilic and the polymer chains display expanded conformation in water, leading to the increasing of the distance between the QDs and the GO sheet. Therefore the FRET process prosess between QDs donor and GO acceptor has been suppressed. In contrast, when the temperature above LCST, the PNIPAM becomes hydrophobic and the distance between QDs and GO sheet decreased. It is worth noting that the the emission was not completely quenched in this aqueous solution, the reason is that the sufficient distance between GO and QDs which modified on the block polymer still provides by the globule state of PNIPAM chains. The reversibility of the thermo-response of QDs/polymer/GO fluorescence nanohybrid was also discussed. As shown in Figure 11b, ten continuous cycles of the efficient on-off switching behaviour happen in the aqueous solution of QDs/polymer/GO fluorescence hybrid. Though cooling the sample solution, the quenched fluorescence could be recovered completely. When re-heated above LCST, the GO based fluorescence nanohybrid performs the similar PL quenching efficiency. The above study indicate that there is a strong interaction between the thermo-responsive block copolymer and GO sheets, which induced the high stability.
